# Multi-Functional Reconfigurable Intelligent Surfaces for Enhanced Sensing and Communication

**DOI:** 10.3390/s23208561

**Published:** 2023-10-18

**Authors:** Khushboo Singh, Mondeep Saikia, Karthick Thiyagarajan, Dushmantha Thalakotuna, Karu Esselle, Sarath Kodagoda

**Affiliations:** 1School of Electrical and Data Engineering, University of Technology Sydney, Sydney, NSW 2007, Australia; dushmantha.thalakotuna@uts.edu.au (D.T.); karu.esselle@uts.edu.au (K.E.); 2Department of Electrical Engineering, Indian Institute of Technology Kanpur, Kanpur 208 016, Uttar Pradesh, India; mondeepsaikia@gmail.com; 3UTS Robotics Institute, University of Technology Sydney, Sydney, NSW 2007, Australia; karthick.thiyagarajan@uts.edu.au (K.T.); sarath.kodagoda@uts.edu.au (S.K.)

**Keywords:** metasurface, metamaterials, reconfigurable intelligent surfaces, intelligent reflecting surfaces, wireless sensors, localization, radar, radar cross section, beam-steering

## Abstract

In this paper, we propose a reconfigurable intelligent surface (RIS) that can dynamically switch the transmission and reflection phase of incident electromagnetic waves in real time to realize the dual-beam or quad-beam and convert the polarization of the transmitted beam. Such surfaces can redirect a wireless signal at will to establish robust connectivity when the designated line-of-sight channel is disturbed, thereby enhancing the performance of wireless communication systems by creating an intelligent radio environment. When integrated with a sensing element, they are integral to performing joint detection and communication functions in future wireless sensor networks. In this work, we first analyze the scattering performance of a reconfigurable unit element and then design a RIS. The dynamic field scattering manipulation capability of the RIS is validated by full-wave electromagnetic simulations to realize six different functions. The scattering characteristics of the proposed unit element, which incorporates two *p-i-n* diodes have been substantiated through practical implementation. This involved the construction of a simple prototype and the subsequent examination of its scattering properties via the free-space measurement method. The obtained transmission and reflection coefficients from the measurements are in agreement with the anticipated outcomes from simulations.

## 1. Introduction

Metamaterials (MMs) and their two-dimensional (2D) counterpart, Metasurfaces (MSs), have gathered an enormous impetus for research and commercialization in the past decade [1]. Several industries have adopted this technology; for instance, Greenerwave launched reconfigurable metasurface-enabled RFID technology in vehicles in 2023 in partnership with an automotive company. Metasurface-assisted satellite communication antennas could be on the market by 2024, and energy-efficient radar systems are being developed in collaboration with Japanese companies AGC and NTT Docomo [2]. Despite the exceptional electromagnetic (EM) wave manipulation capability, the MMs and MSs, once designed, have fixed functionalities that cannot be altered in real-time [3].

The paradigm shift in modern communication systems and future wireless networks seeks post-fabrication tunability and reconfigurability for customized and unique functionalities. Hence, metasurfaces with real-time control over the incident EM field are highly desirable, which has led to a massive surge in the exploration of reconfigurable intelligent surfaces (RISs) [4]. The RIS technology is closely related to MSs and extends the concept further by incorporating the ability to actively modify their properties, allowing for dynamic adaptability to changing communication requirements.

RISs are electrically large, programmable two-dimensional (2D) EM surfaces composed of nearly passive and tunable sub-wavelength elements that facilitate a controlled propagation of information signals between a transmitter and a receiver in a dynamic and goal-oriented way [5,6]. They are envisioned as the key enabler for 6G (beyond 5G) networks that will support driver-less and collaborative transportation, joint communication, localization and sensing, e-health, and tactile internet [7,8]. By altering the parameters of propagating signals, such as phase, amplitude, polarization, frequency and direction, RISs can actively customize the radio environment, creating favorable signal propagation conditions. This will potentially open up new possibilities for enhancing the key performance indicators (KPIs) of wireless communication systems, including spectral and energy efficiency, sensing capability and accuracy, network capacity, and coverage [9].

RIS is emerging as a superior transmission technology to support the transformative and revolutionary goals of 6th-generation wireless communication systems compared to traditional phased arrays, multi-antenna transmitters, and relays. Despite requiring the largest number of scattering unit elements they are comparatively affordable since each scatterer is backed by fewer and cheap components. Thus, RIS is a promising software-defined architecture that is size, weight, and power (SWaP) efficient and has the ability to perform nearly passive beam-forming [10]. The concept of integrating sensing and communication as a solution to tackle the escalating problems arising from spectrum congestion has gained substantial momentum, particularly with the advent of RIS technology. This synergy enables the coexistence of radar-based sensing and wireless communication, allowing them to utilize shared resources like spectrum, RF front-end hardware configurations, and signal processing frameworks, which further improves communication through ubiquitous wireless connectivity and sensing functionalities through connected intelligence within a network [11].

Accurate Wi-Fi-based RF sensing requires a collaborative effort among multiple WiFi access points to measure the influence of a target on the WiFi signal properties, including signal strength, phase, and Doppler shift. On the other hand, mmWave radar-type sensing uses directional beams where the receivers can detect the reflected beams to sense the target. In both Wi-Fi-based RF sensing and mmWave radar, the sensing accuracy is limited by channel conditions. The deployment of RIS can create a virtual line-of-sight (LoS), and hence provide an extra degree of freedom for optimization without deteriorating network performance, which can help in achieving a satisfactory trade-off between the inherently conflicting requirements of communication and sensing [12]. A RIS-aided RF sensing system can achieve high-resolution contact/contact-less sensing by strategically tailoring the radio environment to amplify the desired responses from targets and mitigate interference. Through the manipulation of the propagation environment, this approach not only improves sensing for targets with clear Line-of-Sight (LoS) propagation but also extends radar sensing capabilities to detect objects in obstructed areas that would typically remain undetected due to shadowing effects [13,14,15].

In this article, we propose a reconfigurable unit cell with two positive-intrinsic-negative (*p-i-n*) diodes to control the transmission, reflection, and polarization of the incident EM wave. We then design a multi-functional RIS by arranging the unit cells in a 2D array. Then, by strategically controlling the states of *p-i-n* diodes in the unit element, the RIS can either convert the polarization of the incident EM wave and transmit it as a single beam, dual-beam, or quad-beam or efficiently reflect the incident beam as a single, dual, or quad-beam. Until recently, only a few research works have addressed the challenges of developing a reconfigurable unit cell that can simultaneously control transmission and reflection phases [16], as well as perform polarization conversion. For instance, in a study by Wu et al. [17], a reconfigurable anisotropic digital coding metasurface loaded with electronically controlled *p-i-n* diodes was proposed. However, this unit cell consisted of five substrate layers, resulting in a thicker and bulkier surface than the proposed unit cell in this work. Similarly, in [18], the authors proposed an electronically reconfigurable unit cell for transmit-reflect-arrays, but the fabrication process was complex due to the presence of via holes, and it did not support polarization conversion.

Additionally, in [19], a transmission and reflection-type unit cell integrated with four *p-i-n* diodes was proposed, leading to an increased overall number of *p-i-n* diodes required to design the RIS. In [20], an intelligent reconfigurable metasurface is presented that exhibits the self-adaptive EM manipulation capability and can perform transmission, reflection, and tunable absorption. However, the unit cell configuration is complex, and each unit element uses a total of twelve *p-i-n* diodes. A recent study in [21] proposed a novel reconfigurable transmission-reflection-integrated coding metasurface capable of transmitting and reflection. However, unlike the unit element presented in this work, which incorporates only two *p-i-n* diodes, the unit element in the mentioned study [21] is integrated with three *p-i-n* diodes. The reconfigurable meta-atom recently presented in [22] presents a challenge to this work by offering reflection, transmission, and polarization conversion capabilities using only one switching diode per element. However, upon closer examination, it becomes evident that although this meta-atom has three functionalities, it is limited to only two switching states, making it unsuitable for implementing crucial functions like beam splitting, beam steering, and multi-beam.

In contrast, our unit cell overcomes this limitation with four switching states, providing greater control over surface currents and enabling a wider range of diverse functionalities. It is worth noting that the unit cell requires the fabrication of metallic vias, which adds complexity and cost compared to the simpler planar configuration of our proposed unit cell. In a practical environment, a RIS will be deployed to enhance sensing and communication capability within existing communication systems [13,14]. Alternatively, the RIS itself can be integrated with sensors like UV sensors or gyroscopes, allowing it to adapt its response based on the collected sensing data, as demonstrated in studies such as [23,24].

Moving forward in this document, Section 2 will delve into the methodology behind designing the reconfigurable unit cell. The subsequent Section 3 will showcase the performance of unit cells that can be independently reconfigured, considering various states of *p-i-n* diodes. The paper also entails the design and simulation of a RIS that serves the purpose of achieving polarization conversion, transmission, and reflection, as elaborated in Section 4. Finally, the document concludes in Section 4 with a comprehensive discussion, evaluation, and a glimpse into potential future avenues for research. The RIS is strategically designed and simulated in order to successfully achieve the tasks of polarization conversion, transmission, and reflection. The scattering behavior of the proposed reconfigurable unit cell is experimentally validated by performing free-space measurements on a simple RIS prototype composed of a 12×12 array of unit cells.

## 2. Methodology

For any practical emerging and future application, the RIS must have high-resolution environmental awareness in order to fulfill interactions between the digital and physical worlds [12]. There are several approaches described in the literature to achieve the mentioned goals [25]. These methods include:Meta-Sensing [11]: this method involves a setup consisting of a transmitter, a receiver, a Reconfigurable Intelligent Surface (RIS), and a target space. The RIS customizes the transmitted signal as it enters the target space, where it is reflected by various objects and then picked up by the receiver. By analyzing the received signals, including both Line-of-Sight (LoS) and reflected links, the receiver can derive meaningful sensing information.MetaRadar [26]: the MetaRadar approach utilizes a transmitter, a receiver, a Multiple-Input-Multiple-Output (MIMO) antenna system, and an RIS. In this setup, the RIS enhances the overall channel quality between the antenna array and the sensing targets. This improvement in the channel conditions facilitates better radar performance, potentially leading to more accurate sensing results.MetaLocalization [27]: this technique is designed for indoor localization using elements like access points (e.g., WiFi), an RIS, and multiple users needing indoor positioning. The RIS actively modifies radio maps, reducing the similarity in received signal strength values between adjacent locations. This aids in improving the precision of indoor positioning.MetaSLAM (Simultaneous Localization and Mapping) [28]: MetaSLAM employs the RIS to enhance the accuracy of the SLAM technique. SLAM involves a system, often robotic, that simultaneously creates a map of its environment and locates itself within the map. The RIS helps in refining the accuracy of this process.

In essence, these methods leverage the capabilities of RIS to enhance various aspects of wireless sensing, radar, localization, and mapping. To address the challenge of spectral congestion, a prudent approach involves adopting an integrated architecture that combines sensing and communication. This integration serves to enhance spectral efficiency [12]. In order to enable such a setup, incorporating a reconfigurable intelligent surface with a dedicated sensing unit becomes essential. Hence, the inclusion of a sensor, along with a microcontroller unit (MCU) and/or a Field Programmable Gate Array (FPGA), is crucial for the real-time management and control of the RIS operations [25].

The operational concept of the sensing system based on the RIS is detailed using a block diagram, as depicted in Figure 1.

The sensing unit detects a change in the environment and converts the sensing data to a digitized information signal, which is then sent to a controller such as a microcontroller unit (MCU) and/or a Field Programmable Gate Array (FPGA). The FPGA is programmed to generate voltage distribution based on the digital code sequence. The voltage distribution controls the biasing of tunable elements in the RIS (such as *p-i-n* diodes or any other tunable RF switches, such as varactor diodes, Micro-Electromechanical-Systems (MEMS), or Nano-Electromechanical-Systems (NEMS)), which enables real-time control of the scattering properties of the RIS.

### 2.1. Unit Cell Design

The operational concept of the sensing system based on the RIS is detailed using a block diagram, as depicted in Figure 1. This study introduces a Reconfigurable Intelligent Surface designed to execute diverse functions, including transmission (single beam, dual beam, or quad-beam with $90^{\circ }$ polarization conversion) and reflection (single beam, dual beam, or quad-beam with the same polarization as the incident electromagnetic signal). The specific functions are determined by configuring the states of the incorporated switches.

To realize such a multi-functional RIS, we investigated a 2-bit transmission and reflection phase programmable unit element that comprises three dielectric layers and four metal layers embedded with two *p-i-n* diodes, as shown in Figure 2. The configuration of this unit cell draws inspiration from the concept of a time-modulated polarization-rotating frequency-selective surface, as proposed in [29]. The top and bottom surfaces of the unit cell consist of slanted metallic gratings positioned in a way that they are perpendicular to each other and are responsible for polarization conversion for transmitted waves. The dielectric layers are all constructed from FR4 epoxy substrate with a relative permittivity of $\epsilon_{r}=4.4$ and a loss tangent of tanδ=0.02. An air gap, with a height denoted as *g*, separates the two substrate layers. At a frequency of 1.8 GHz, the value of *g* is set to 15 mm. Both the top and bottom substrate layers share the same thickness of t=0.5 mm. However, the middle layer is slightly thinner, with a thickness denoted as $t_{1}$, which is 0.3 mm. Each metallic layer on the top and bottom of the middle substrate incorporates a *p-i-n* diode. Specifically, the design employs the Infineon BAR63-02V *p-i-n* diode [30]. To include the *p-i-n* diodes in the unit cell simulation, we modeled them as equivalent RLC circuits as shown in Figure 2f,g.

### 2.2. Unit Cell Simulations

The full-wave EM simulations are carried out using the commercial EM solver CST Microwave Studio. For the *p-i-n* diodes, the inductance is maintained at a constant value of L=0.6 nH. However, the resistance changes based on variations in the bias voltage. In the “ON” state (i.e., when the bias voltage surpasses the threshold voltage), the *p-i-n* diode is represented as a series RLC circuit with a minimal resistance of *R* = 1 Ω and inductance L=0.6 nH. Conversely, in the “OFF” state, the *p-i-n* diode can be approximated as a parallel RLC circuit featuring a higher resistance of R = 5 kΩ, a capacitance around C=0.25 pF, and an inductance L=0.6 nH in series.

Figure 3a illustrates the arrangement of the *p-i-n* diodes positioned on the two sides of the middle substrate layer within the unit cell. The diode situated on the upper side of the middle layer is labeled as D1, while the diode on the lower side is labeled as D2. Depending on the ON and OFF states of these two *p-i-n* diodes, there exist a total of four distinct switching state combinations, as presented in Figure 3b. To simulate the unit cell, we implement periodic boundary conditions along with Floquet port excitation [31,32]. The polarization (independent of scan angle) of the incident Floquet modes is $45^{\circ }$, such that the fundamental TE(00) and TM(00) modes are linearly combined to generate the first mode with polarization aligned to $45^{\circ }$ with respect to the *u-axis* of the Floquet port (in CST MWS).

Notably, it is important to emphasize that when both diodes are in the ON state or when both are in the OFF state, the unit cells exhibit significant reflection characteristics. On the other hand, when one diode is ON while the other is OFF, the unit cells exhibit substantial transmission attributes. Furthermore, in the case of transmission, it is observed that the output electric field’s polarization is orthogonal to the polarization of the incident electric field. Conversely, the reflected electric field maintains the same polarization as the incident electric field. The polarization conversion achieved in this unit cell depends entirely on the switching states of the PIN diodes in the middle layer.

To delve into the operational concept of the switchable polarization-rotating unit cell, let us consider two coordinate systems, namely, XYZ and UVZ. In this configuration, the UV-axis is oriented at a $45^{\circ }$ angle relative to the XY-axis, as illustrated in Figure 2a. The top layer of the unit cell features copper strips aligned along the U-axis, while the bottom layer is equipped with copper strips aligned along the V-axis. On the intermediate switching layer, PIN diodes are embedded within metal strips along the X-axis on the top surface and along the Y-axis on the bottom surface. This unit cell exhibits polarization rotation properties in two distinct switching states: 01 and 10. To gain insight into the behaviors of the unit cell, we conducted separate simulations for each layer. In the top layer, which essentially functions as a wire grid polarizer, it obstructs U-polarized waves while allowing V-polarized waves to pass through. Conversely, the bottom layer permits the passage of U-polarized waves while inhibiting V-polarized waves. Considering the incidence from the bottom, in the 10-switching state, the middle layer converts the incident U-polarized wave into four components: a reflected U-polarized wave, a reflected V-polarized wave, a transmitted U-polarized wave, and a transmitted V-polarized wave. Consequently, when all the layers are assembled together, the overall magnitude of the transmitted V-polarized wave increases due to multiple reflections between the layers of the unit cell. A similar explanation applies to the 01-switching state.

The simulated transmission and reflection coefficients of the unit cell are plotted in Figure 4. The transmission magnitudes and phases are plotted in Figure 4a and Figure 4b, respectively, for the two different switching state combinations: (1) when D1 is OFF and D2 is ON, and (2) when D1 is ON and D2 is OFF. We observe that the transmission magnitudes for the two switching states are nearly identical and higher than −3 dB from 1.35 GHz to 2.1 GHz. The similarity in transmission magnitudes can be attributed to the fact that the overall network configuration of the unit cell remains unchanged when transitioning between the 01 and 10 switching states (i.e., from D1 = OFF and D2 = ON to D1 = ON and D2 = OFF), except for the change in the diode that is turned ON which causes a reversal in the electric field direction. The resulting change in the electric field direction creates a difference of $180^{\circ }\pm 0.5^{\circ }$ in the transmission phases of the two states (01 and 10) as can be seen in Figure 4c.

The reflection magnitudes and phases are plotted in Figure 4d and Figure 4e, respectively, for two different switching state combinations; (1) when (D1 = OFF, D2 = OFF) and (2) when (D1 = ON and D2 = ON). The reflection magnitude is higher than −2.2 dB for both switching state combinations from 1 GHz to 3 GHz, whereas the difference between the reflection phases increases gradually with frequency increasing from 1 GHz to 3 GHz. However, if we consider a phase range of $180^{\circ }\pm 30^{\circ }$, we can still obtain a narrow band operation from 1.44 GHz to 1.64 GHz, as is shown in Figure 4f. Such variations in phase are acceptable to modulate the reflecting field from the RIS [24].

### 2.3. Multi-Functional RIS Design and Simulations

Based on the investigations made on the scattering performance of the unit cell, we develop a 20×20 multi-functional RIS with a total size of 640×640 mm${}^{2}$, that can be dynamically programmed to achieve different responses. Thus, there are a total of 800 *p-i-n* diodes. The ON state of the *p-i-n* diode corresponds to the digital code 0, and the OFF state corresponds to the digital code 1. For simulation simplicity, we divide the surface into a 2×2 sub-region, with each sub-region having 10×10 unit cells. In this case, we assume that all sub-regions can be biased independently (a biasing scheme has been proposed later in Section 3), and the modulation for scattering can be realized. However, in practice, each element of the RIS can be controlled independently, and more precise control is possible. Or the *p-i-n* diodes in the same column may share the same bias line to consume the same DC voltage and work under the same conditions.

We simulated six different configurations to validate the multi-functional capability of the proposed RIS. The plane-wave excitation using waveguide port and open boundary conditions have been applied for the full RIS simulations in CST microwave studio. The coding sequence and the corresponding 3-D radiation patterns for transmitting RIS are shown in Figure 5 and for reflecting RIS in Figure 6. In practice, we will implement the coding sequence using FPGA. FPGA will generate a digital code sequence corresponding to the desired scattering pattern to exercise such control over the EM response of the proposed RIS, which will then distribute the voltages to the bias lines and operate the RIS to realize different functions. Each state combination is depicted using different colors for clarity. Theoretically, the beam deflection angle of a coding metasurface composed of N×N elements is as follows [23].
(1)$$\begin{array}{c} \theta =sin^{-1}\left(\frac{\lambda }{nP}\right) \end{array}$$
where λ is the wavelength at the central frequency, *n* is the number of units in each period, and *P* is the periodic size of each unit. The deflection of the beam after splitting can be calculated by using Equation (Equation 1). We used the proposed RIS to achieve specular transmission, specular reflection, and beam splitting (for both transmission and reflection modes). We observe that the directivity is higher in for transmission (Figure 5), compared to the reflection states (Figure 6). This decline is attributable to the fact that the transmission magnitude of the proposed unit cell is higher (exceeding −1.5 dB), whereas the reflection magnitude is below −1.6 dB, as can be seen in Figure 4. Consequently, in the reflection configuration, due to spurious scattering, the overall magnitude of the reflected signal experiences a reduction.

## 3. Experimental Verification

A basic prototype was fabricated using an array of 12 × 12 unit elements, as depicted in Figure 7a, to empirically verify the scattering properties of the suggested unit element. The substrate layers were assembled using pillars of Rohacell foam, which was employed to accommodate the air gap within the unit cell. The electrical characteristics of Rohacell foam closely resemble those of air, featuring a permittivity of 1.050 and an exceptionally low loss tangent of less than 0.00002. The *p-i-n* diode is soldered within the space between the two metal strips present in the middle substrate layer, on both faces. The schematic illustrating the biasing arrangement for the *p-i-n* diodes on the upper and lower surfaces of the middle substrate board is shown in Figure 7b,c.

The *p-i-n* diodes receive biasing voltages through a four-channel DC power supply.

The magnitudes of the co-polarized reflection coefficient and the cross-polarized transmission coefficients are measured for four distinct switching states using the free space measurement technique [33]. In this measurement setup, a set of horn antennas is positioned at opposite ends of the prototype within an anechoic chamber as illustrated in Figure 8.

The horn antennas are positioned at a predetermined distance such that the prototype is illuminated by a plane wave. The prototype is fixed using an absorbing foam window. The width and height of the prototype must be sufficiently larger than the horn antennas to avoid inaccuracy caused by signal diffraction at sample edges. By performing the measurements on a 12×12 array of unit elements, as opposed to a single element, the obtained results become more accurate and close to the predictions made by electromagnetic (EM) simulations assuming periodic boundary conditions in both the *x* and *y* directions. Replicating the unit elements in both *x* and *y* directions multiple times to create a larger aperture mimics the assumed periodicity during the EM simulation. This measurement process is undertaken to verify the scattering characteristics of the suggested unit cell, particularly in relation to the various switching state combinations of the two diodes. Figure 9 depicts the observed magnitudes of cross-polar transmission and co-polar reflection coefficients. These measurement outcomes align with the information presented in Figure 4b, which illustrates the list of switching states for the *p-i-n* diodes and the corresponding scattering characteristics of the unit cell, as predicted through comprehensive full-wave CST simulations. Consequently, this validation process affirms the accuracy of the unit cell’s scattering behavior as proposed.

## 4. Conclusions and Future Work

We developed a versatile reconfigurable unit element that can convert polarization, transmit, and reflect signals. This empowers us to control how the scattered signals behave in real-time when the proposed reconfigurable surface is integrated with detection, artificial neural networks, and feedback systems. Using the proposed reconfigurable surface, we can establish a smart platform that autonomously manipulates the interactions between waves, information, and matter. Due to the dynamic capability of our unit element to efficiently transmit as well as reflect incident electromagnetic signals, they can also be harnessed for the design of Simultaneously Transmitting and Reflecting (STAR) RISs [34]. The STAR RIS structures divide incoming wireless signals into transmitted and reflected ones, influencing signal propagation throughout the surrounding space. This methodology will also enhance existing techniques for spatial field reconstruction [35] and estimation [36] for environmental monitoring indoors.

Additionally, it will bolster temporal monitoring of sensors operating in hostile environments [37,38], wireless robot localization [39], soft-robotics [40] and SLAM [41] strategies for complex environment exploration.

In the upcoming stages of our project, our objective is to seamlessly embed a detection module (e.g., power detector, inertial measurement unit (IMUs) sensors or gyroscope) to detect the direction/angle of arrival in MetaRadar applications and an FPGA control platform, into the surface. This integration will enable on-surface signal processing and reconfiguration in response to changing conditions or specific needs. Incorporating a detection module and FPGA control platform directly into the surface streamlines signal processing and reconfiguration, enhancing the system’s adaptability and performance. This integrated approach eliminates the requirement for external processing units, reducing latency and simplifying deployment and maintenance. To implement the proposed reconfigurable surface for MetaSLAM, a combination of various sensors and algorithms will function as a unified intelligence framework. Additional information about the types of detection and algorithms that can be employed is readily available in [25].

Our proposed unit element’s design is notably straightforward and planar, making fabrication easy compared to other approaches with similar capabilities of reflection and transmission [18,20]. This compilation serves as a comprehensive resource for understanding the design concepts of reconfigurable intelligent surfaces. It targets both academic and industry researchers, along with research organizations, aiding in developing a strong foundational understanding and informed decision-making for future wireless system designs involving reconfigurable surfaces.

## Figures and Tables

**Figure 1 sensors-23-08561-f001:**
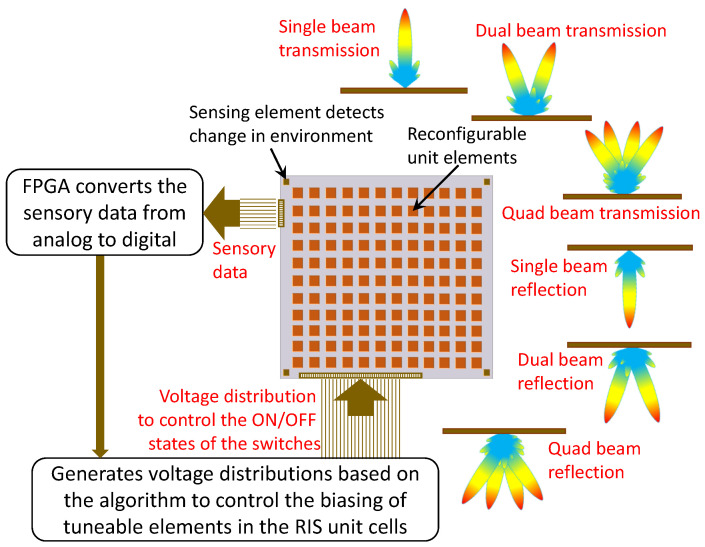
Schematic of a RIS-based sensing system. The RIS offers various EM functions, including the transmission of single, dual, or quad beams, the reflection of single, dual, or quad beams, polarization conversion, and many more, depending on the states of the tunable switches in the RIS.

**Figure 2 sensors-23-08561-f002:**
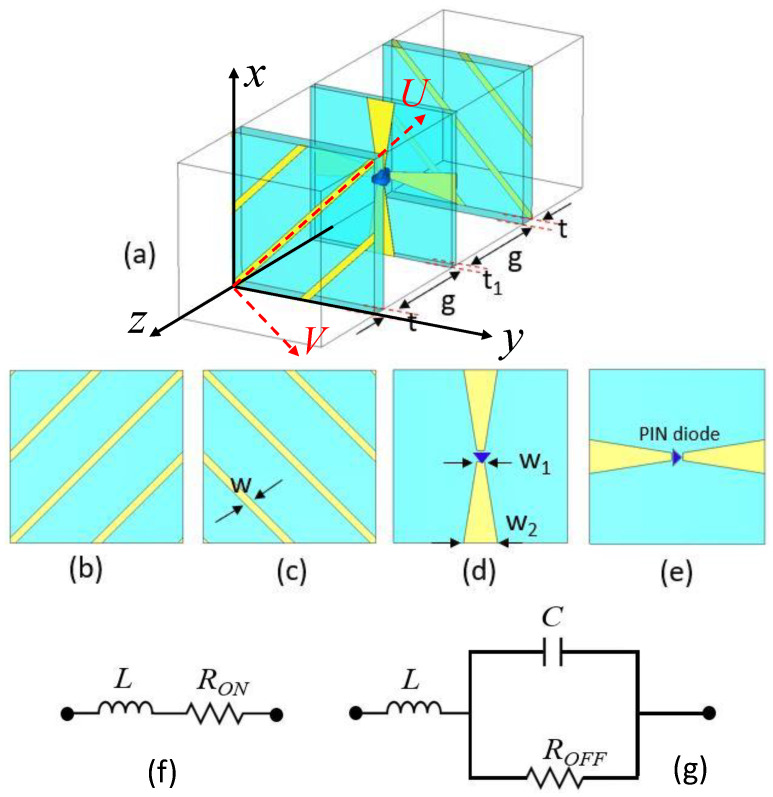
The unit cell model for the proposed RIS is displayed here in (**a**) perspective view showing three dielectric substrates and four metal layers (an air gap separates the substrates), (**b**) top view of the top layer, (**c**) the bottom view of the bottom layer, (**d**) the top view of the middle layer, and (**e**) the bottom view of the middle layer. Equivalent circuit model for Infineon BAR63-02V *p-i-n* diode: (**f**) ON and (**g**) OFF states.

**Figure 3 sensors-23-08561-f003:**
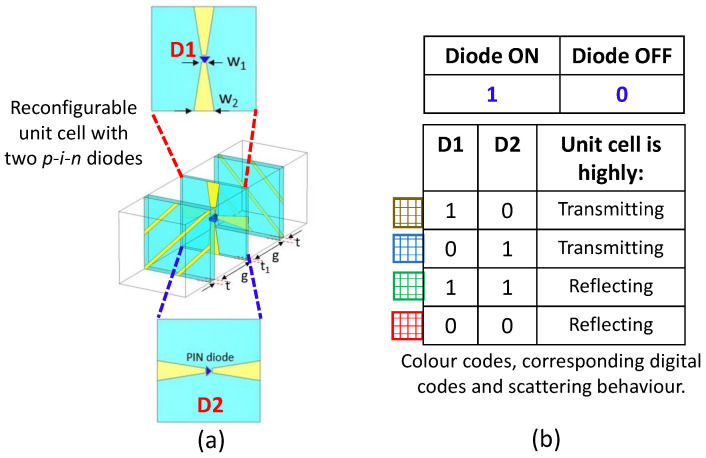
(**a**) Exploded view of the middle layer of the unit cell showing the orientation of *p-i-n* diode on both sides. (**b**) Table listing the switching states of the *p-i-n* diodes and corresponding scattering behavior of the unit cell.

**Figure 4 sensors-23-08561-f004:**
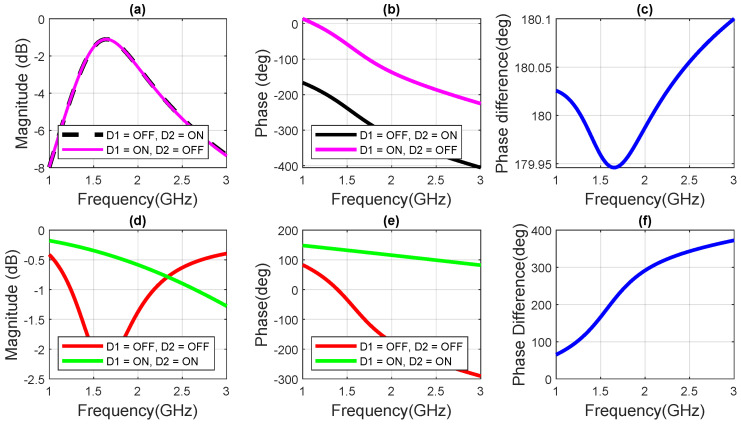
The scattering response of the proposed unit cell for different switching states. (**a**) The transmission magnitudes and (**b**) the transmission phases, of the cross-polar component when either of the *p-i-n* diodes is ON. (**c**) The difference between the transmission phases of the two transmitting unit cells. (**d**) The reflection magnitudes and (**e**) the reflection phases, of the co-polar components when both *p-i-n* diodes are either ON or OFF. (**f**) The reflection phase difference between the two reflecting unit cells.

**Figure 5 sensors-23-08561-f005:**
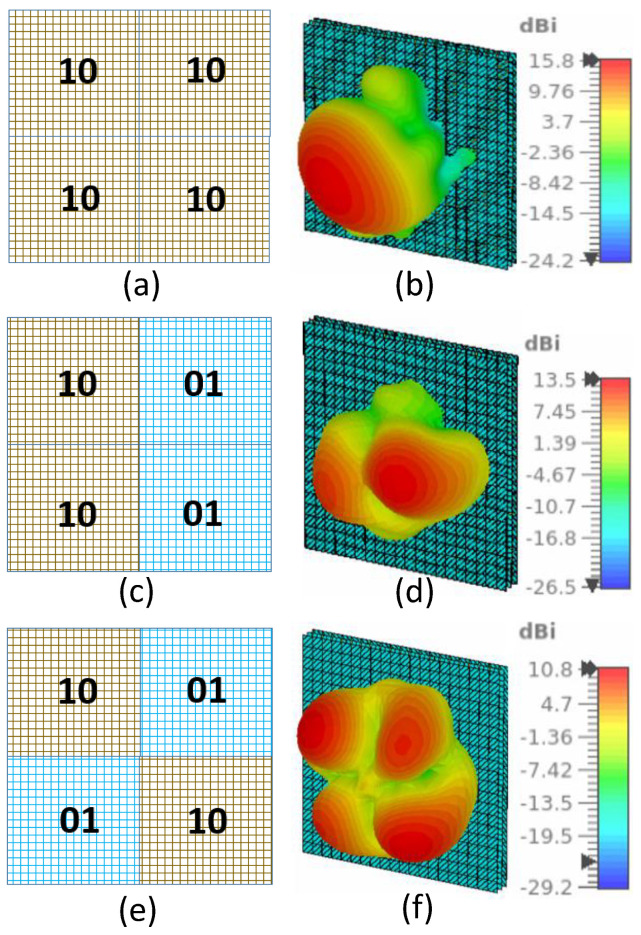
RIS configurations and corresponding three-dimensional (3-D) transmitted radiation patterns. (**a**) RIS with (D1 = ON; D2 = OFF) for all four sub-regions and (**b**) the corresponding 3-D radiation pattern. (**c**) RIS with (D1 = ON; D2 = OFF) for two adjacent sub-regions and (D1 = OFF; D2 = ON), for the other two adjacent sub-regions and (**d**) the corresponding 3-D radiation pattern. (**e**) RIS with (D1 = ON; D2 = OFF) for alternate sub-regions and (D1 = OFF; D2 = ON), for the other two alternate sub-regions and (**f**) the corresponding 3-D radiation pattern.

**Figure 6 sensors-23-08561-f006:**
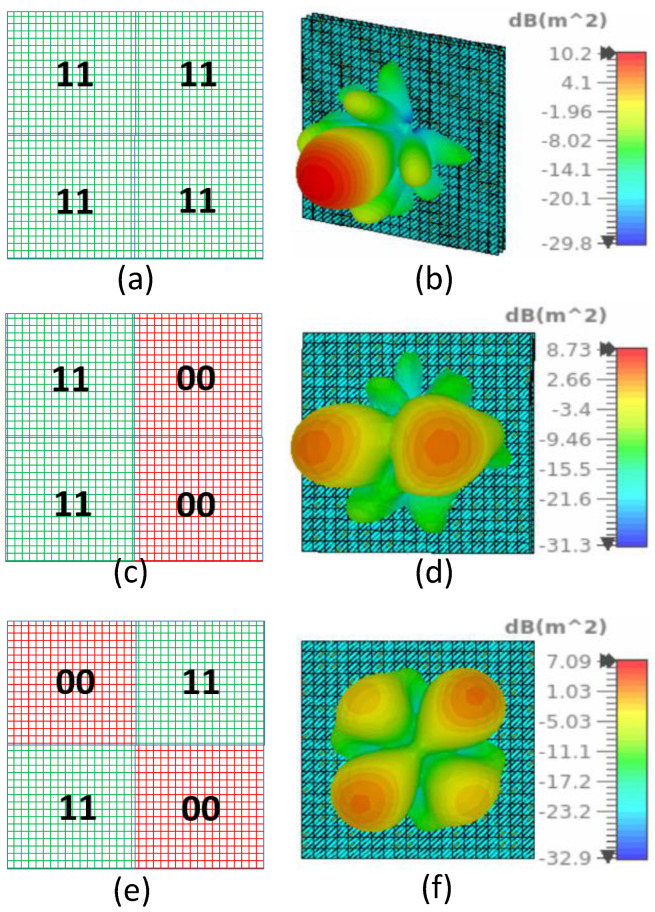
RIS configurations and the corresponding 3-D bistatic retro-directive radiation pattern. (**a**) RIS with (D1 = ON; D2 = ON) for all four sub-regions and (**b**) the corresponding 3-D bistatic radiation pattern. (**c**) RIS with (D1 = ON; D2 = ON) for two adjacent sub-regions and (D1 = OFF; D2 = OFF), for the other two adjacent sub-regions and (**d**) the corresponding 3-D bistatic radiation pattern. (**e**) RIS with (D1 = OFF; D2 = OFF) for alternate sub-regions and (D1 = ON; D2 = ON), for the other two alternate sub-regions and (**f**) the corresponding 3-D bistatic radiation pattern.

**Figure 7 sensors-23-08561-f007:**
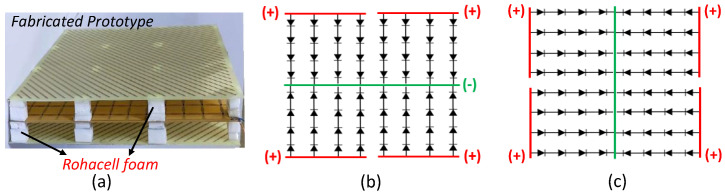
(**a**) Side-view of the fabricated prototype of the proposed RIS. (**b**) Biasing scheme for diodes on the top surface of the middle substrate layer and (**c**) Biasing scheme for diodes on the bottom surface of the middle substrate layer.

**Figure 8 sensors-23-08561-f008:**
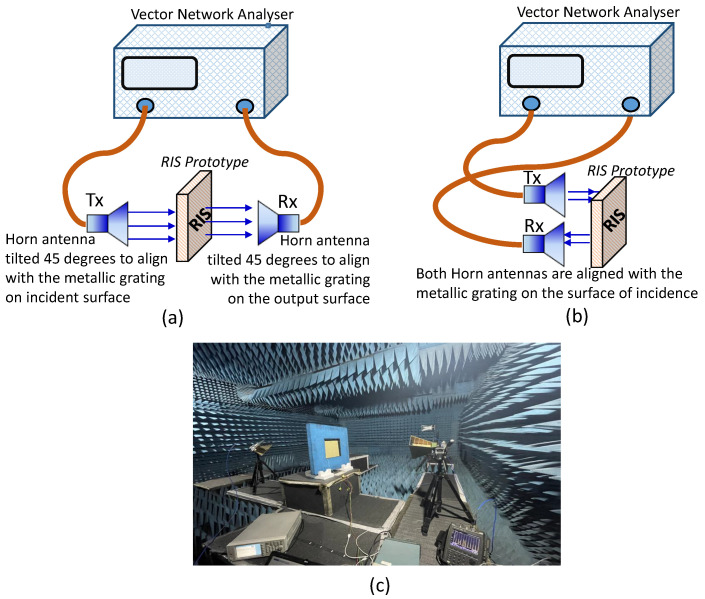
Schematic for the free space measurement set-up in the anechoic chamber to validate the scattering behavior of the proposed reconfigurable unit cell embedded with two *p-i-n* diodes. The transmitter (Tx) and receiver (Rx) antennas are rotated accordingly to account for the variation in polarization. (**a**) Measuring the transmission coefficients. (**b**) Measuring the reflection coefficients. (**c**) Actual measurement setup in the anechoic chamber.

**Figure 9 sensors-23-08561-f009:**
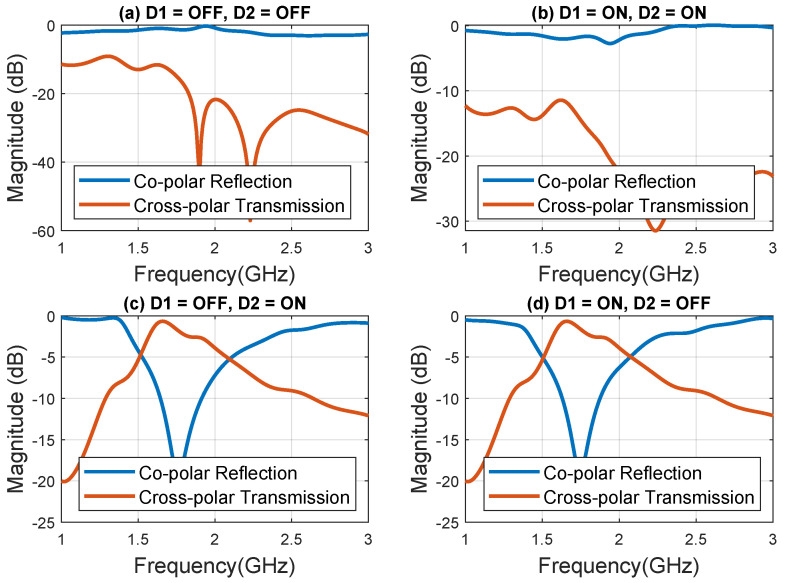
Measured magnitude for cross-polar transmission and co-polar reflection coefficients of the fabricated prototype for different states of the *p-i-n* diodes (**a**) 00 state, (**b**) 11 state, (**c**) 01 state, and (**d**) 10 state.

## Data Availability

Not applicable.

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
