# Peer review of "Multi-Functional Reconfigurable Intelligent Surfaces for Enhanced Sensing and Communication"

_sensors, 2023, doi:10.3390/s23208561_

Round 1

Reviewer 1 Report

In fact, this work has average quality and needs improvement in some sections. 

1-  The author's work is similar to previous studies  such as 

"Mechanically Tunable Terahertz Circular Polarizer with Versatile Functions"

As the first point, I think the author should highlight these works and mention to quality of the slant metasurface for polarization control, and clarify why the use of two polarizer layers, it look like it is converted from linear to circular and circular to linear.

2-The transmission matrix and equivalent circuit model can help to understand physics beyond it as  presented in "Mechanically Tunable Terahertz Circular Polarizer with Versatile Functions"

3- The introduction sentences are too long and boring to read without adding any special information to the readers.

4- The difference between the various switching mode is not too much why and please describe  what is your main goal for design it 

Author Response

We extend our gratitude to the reviewer for their time, effort, and constructive feedback, which has significantly enhanced the readability and overall quality of this paper. An elaborate explanation and is provided as an answer to each of the reviewer's comments in the document attached.

Reviewer 2 Report

The authors introduce a reconfigurable intelligent surface (RIS) with the ability to dynamically alter electromagnetic wave properties for dual-beam or quad-beam transmission and polarization conversion, enhancing wireless connectivity in challenging line-of-sight scenarios. When combined with sensors, RIS systems can perform joint detection and communication in wireless sensor networks. The research validates the RIS's dynamic field manipulation capabilities through electromagnetic simulations and practical implementation, confirming alignment between measured transmission and reflection coefficients and simulated expectations. This is definitely worthy of consideration. However, there are some technical issues that need further clarification before this paper can be accepted.

1. The author's presentation of certain conclusions in the study exhibits instances of overstatement. For instance, the authors assert the categorization of their devices as "Intelligent Surfaces," yet the experimental verification lacks any demonstrable algorithmic augmentation. In essence, the author's presentation is more indicative of "Reconfigurable Surfaces." While it is indisputable that the methodology posited by the author holds the potential for advancement toward achieving Intelligent Surfaces, the absence of substantiating empirical data is conspicuous. These assertions seem to primarily reflect the author's conjecture and may be considered somewhat incongruous with the study's overarching claims. Consequently, it is recommended that the author revisits and revises the concluding statements in light of these observations.

2. Similar issues manifest in relation to assertions pertaining to augmented sensing and communication capabilities. The title posits the utilization of these surfaces for the purpose of enhancing sensing and communication. However, it is imperative to underscore that these proclamations are merely conjectural, lacking empirical substantiation. Furthermore, within the authors' discourse, the potential utility of these surfaces for heightened sensing is articulated. In light of this, it is incumbent upon the author to provide a comprehensive enumeration of the specific sensing components in the "Conclusion and Future Work" section. This would serve the dual purpose of addressing lingering uncertainties and illuminating the practical applications of this technology. For example, what is it used to detect? What is the detection limit? What improvements are there over existing detection technologies?

3. In Figure 1, the step "FPGA converts sensing data from analog to digital" is repeated redundantly. A single instance of this conversion suffices.

Author Response

We express our appreciation to the reviewer for dedicating their time and effort, as well as for their constructive reviews that have contributed to enhancing the readability and overall quality of this paper. The detailed response to each comment is provided in the attached document.

Reviewer 3 Report

This paper presents a reconfigurable intelligent surface (RIS) design consisting of a tunable unit cell that can control the transmission, reflection, and polarization of incident electromagnetic waves. The contents are good, but I still have several quick questions.

1. Can the authors also show the magnitude/phase response when the D1/D2 is configured as ON/OFF or OFF/ON?

2. It would be better if the authors could show several photos of the experiment setups, like the wiring between the FPGA and the RIS, and the measurement setup.

3. In Fig. 5, the units are dBi, while the units are dB in Fig. 6. The units need to be consistent, Additionally, for the transmitting case (Fig. 5), the minimum magnitude is 10.8 dBi, which is almost the same as the reflecting case that shown in Fig. 6(b). Why is the magnitude of the transmitting case almost the same as the reflecting array?

4. Can the authors also show the characterized radiation pattern of the fabricated metasurfaces?

Author Response

We would like to convey our gratitude to the reviewer for investing their time and effort, and for their valuable feedback that has played a pivotal role in improving the readability and overall quality of this paper. A comprehensive response to each comment can be found in the attached document.

Round 2

Reviewer 1 Report

the paper can be published in the current form 

Reviewer 2 Report

The authors have addressed my concerns. I recommend the acceptance of this manuscript.